# Sulforaphane Impact on Reactive Oxygen Species (ROS) in Bladder Carcinoma

**DOI:** 10.3390/ijms22115938

**Published:** 2021-05-31

**Authors:** Hui Xie, Felix K.-H. Chun, Jochen Rutz, Roman A. Blaheta

**Affiliations:** Department of Urology, Goethe-University, 60590 Frankfurt am Main, Germany; xiehui0831@outlook.com (H.X.); Felix.Chun@kgu.de (F.K.-H.C.); Jochen.Rutz@kgu.de (J.R.)

**Keywords:** bladder cancer, sulforaphane, ROS, oxidative stress, Keap1-Nrf2

## Abstract

Sulforaphane (SFN) is a natural glucosinolate found in cruciferous vegetables that acts as a chemopreventive agent, but its mechanism of action is not clear. Due to antioxidative mechanisms being thought central in preventing cancer progression, SFN could play a role in oxidative processes. Since redox imbalance with increased levels of reactive oxygen species (ROS) is involved in the initiation and progression of bladder cancer, this mechanism might be involved when chemoresistance occurs. This review summarizes current understanding regarding the influence of SFN on ROS and ROS-related pathways and appraises a possible role of SFN in bladder cancer treatment.

## 1. Introduction

Bladder cancer is the 10th most common form of cancer worldwide, with an estimated 549,000 new cases and 200,000 deaths per year [1]. High recurrence rates and progression to higher tumor stages are characteristic. Therapeutic strategy depends on the histologic subtype, broadly categorized into non-muscle invasive, muscle invasive or metastatic bladder cancer. Transurethral resection combined with follow-up intravesical chemotherapy or Bacillus Calmette–Guerin (BCG) immunotherapy are the main treatment approaches for non-muscle-invasive bladder cancer [2]. Still, patient outcome remains unsatisfactory with a 5 year recurrence rate ranging from 31% to 78% and a progression rate of up to 45% [3]. When the cancer has invaded the muscularis propria of the bladder wall (stage T2a and T2b), perivesical fat (T3a and T3b), adjacent organs (T4a) or the pelvic or abdominal wall (T4b) with rapid development of systemic micrometastases, treatment becomes a severe therapeutic challenge. Here, the treatments of choice are cisplatin-based combination chemotherapy followed by radical cystectomy and bilateral pelvic lymphadenectomy. Urinary diversion is considered. However, in the majority of patients with metastatic bladder cancer, this treatment is palliative.

Dissatisfactory response in terms of acquired resistance, along with strong side effects caused by conventional tumor treatment, drive many cancer patients to seek complementary alternative medicine (CAM) options. Worldwide, approximately 50% of cancer patients are reported to use CAM, with variations depending on the tumor entity, patient socio-demographics and country [4]. The oral use of natural herbs and plant compounds is one of the most commonly applied CAM options [5]. Indeed, clinical studies indicate that incorporating plant derived compounds into conventional antitumor protocols may improve the success of bladder cancer treatment [6].

The present review focuses on the use of the phytochemical drug, sulforaphane (SFN), to treat bladder cancer. The precursor of SFN, glucoraphanin, is highly enriched in cruciferous vegetables from the Brassicaceae family, including broccoli, cauliflower and cabbage. The mean content of glucoraphanin per gram uncooked broccoli has been calculated at 0.38 μmol, and levels of glucoraphanin evaluated in over 75 different genotypes of field-grown hybrid broccoli average between 0.88 μmol and 1.10 μmol glucoraphanin per gram fresh weight [7]. Importantly, glucoraphanin itself is not bioactive. Rather, enzymatic hydrolysis by myrosinase, present in the plant tissue or in the mammalian microbiome, is necessary to form the active component, SFN.

Ten years ago, a redox regulating function of SFN was identified [8,9]. Disrupting the redox balance by increasing free radicals, predominantly reactive oxygen species (ROS), has been shown to be closely associated with carcinogenesis and metastatic progression [10]. The degree and direction of disruption may be crucial since a moderate increase of ROS has been associated with enhanced cancer progression, while a strong and massive increase of ROS may act as a cancer suppressor by inducing apoptosis [10]. Based on in vitro, in vivo and patient studies, SFN has been shown to exert both chemopreventive and tumor-suppressive properties through multifaceted mechanisms [11]. The current review deals with the question of how SFN may modulate ROS and how intervention into ROS-related signaling through SFN may provide innovative antitumor strategies.

## 2. ROS and Cancer

ROS refers to a collective term of oxygen-containing species, divided into radicals such as superoxide (O_2_▪−), hydroxyl radical (OH ▪), peroxyl (RO_2_ ▪), alkoxyl (RO ▪) hydroperoxyl (HO_2_▪) and non-radicals, such as hydrogen peroxide (H_2_O_2_), hypochlorous acid (HOCl) and ozone (O_3_), among others [12]. Intracellular processes as well as exogenous factors are both relevant for ROS production. Endogenously, ROS is generated as a byproduct of the cytochrome P-450 reaction cycle in subcellular organelles such as mitochondria and peroxisomes. Exogenously, ROS is produced via the environment through pollutants, tobacco, smoke, drugs, xenobiotics, radiation and other mediators [13]. One of the major endogenous ROS generators, the NADPH oxidase (NOX) protein family, includes seven distinct isoforms: NOX1–NOX5, dual oxidase (DUOX) 1 and DUOX2 [14]. The process of ROS production via the NOX family serves to transport electrons across biological membranes, reducing oxygen to superoxide [15].

In healthy tissue, ROS is exactly balanced by multiple antioxidants to guarantee redox homeostasis. Overexpression of ROS, due to an imbalanced redox system, is associated with cancer initiation, progression and metastasis, and it has been postulated that excessive oxidative reactions lead to DNA, protein and lipid damage [16]. NOX overexpression has been found in pre-malignant lesions and various malignant cancers [17,18,19,20]. Increased growth activity of multiple tumor types, including pancreas, breast, lung, colon, prostate and bladder cancer, has been traced back to massive ROS generation [13]. Chronic inflammation plays a role in the initiation of carcinogenesis, since free radicals are generated [21]. Here immune cells are recruited to the damage site and a ‘respiratory burst’ with increased oxygen intake and a dramatic release of ROS [22] ensues. ROS generation in turn triggers the recruitment of further inflammatory cells to secrete pro-inflammatory factors, causing secondary damage amplification [21]. A vicious circle is established, forcing the initiation and progression of cancer [23]. Hence, therapeutic strategies have been suggested for eliminating ROS to restore the redox balance and stop tumor growth.

Still, the relevance of ROS in cancer development is controversial since overproduction of ROS is also regarded as toxic to cancer cells. Sufficient evidence indicates that ROS also promotes anti-tumorigenic signaling that is evoked by oxidative stress-induced tumor cell death. In pancreatic and breast cancer cells an elevated ROS level has been shown to increase caspase-8, -9 and -3, leading to apoptosis [24]. Feng and co-authors have revealed that increased ROS accumulation may be one of the pivotal mechanisms leading to the inhibition of cell proliferation, migration and invasion of human osteosarcoma cells [25]. Recent work points to beneficial effects of ROS in patients with chronic lymphocytic leukemia. Here H_2_O_2_ accumulation was associated with a favorable clinical outcome [26]. In a comparative study, a high level of ROS inhibited the development and progression of hepatocellular cancer [27].

These findings coincide with observations regarding radiotherapy and chemotherapy. Both strategies induce oxidative stress and ROS-mediated cell damage in cancer cells by increasing ROS above a critical threshold, necessary to initiate apoptosis [28,29,30]. Thus, though an intracellular increase of ROS may force tumor growth and progression, excessive ROS levels may become toxic to the cells [31]. ROS therefore displays both pro- and anti-tumorigenic properties, whereby a moderate ROS-increase exerting carcinogenic properties must be distinguished from very strong ROS accumulation causing cell damage.

ROS affects cell gene expression through various pathways. Classically, ROS modulation provokes distinct alteration in the nuclear factor erythroid-derived 2-like 2 (Nrf2) pathway (Figure 1) with Nrf2 serving as the key transcription factor activated by oxidative stress. Under normal, unstressed conditions, Nrf2 is located in the cytoplasm, complexed to the Kelch-like ECH-associated protein-1 (Keap1). With increasing ROS, Keap1 undergoes a conformational change, triggering the escape of Nrf2 from the protein complex and subsequent transfer to the antioxidant response element (ARE) in the nucleus. Binding of Nrf2 along with the transcription factor s-Maf to ARE then causes transcription of downstream genes and expression of enzymes responsible for antioxidant effects such as heme oxygenase-1 (HO-1) and NADPH quinone oxidoreductase-1 (NQO-1) [32,33].

Nrf2 activation has been observed in various cancer types, protecting cancer cells from oxidant induced cytotoxicity [34]. This may explain why increased Nrf2 in the nucleus has been shown to enhance growth and metastasis of breast cancer cells [35]. Nrf2 has also been shown to promote esophageal cancer cell proliferation via detoxification of ROS [36]. In human hepatocellular carcinoma enhanced Nrf2 is considered a poor prognostic marker [37].

Still, the role of Nrf2 in tumorigenesis is ambiguous. Induction of HO-1 and NQO-1 by the Nrf2 pathway may prevent lung tumorigenesis by attenuating ROS mediated inflammation [38]. Liu et al. has observed that upregulated Nrf2 sensitized prostate cancer cells to radiation via decreased basal ROS levels [39]. A breast cancer study has revealed that Nrf2 activation may inhibit the invasiveness and metastatic potential of the tumor via downregulation of hypoxia-inducible factor 1α (HIF-1α). In fact, HIF-1α has been positively correlated to the ROS level and is suggested to play a crucial role in facilitating tumor progression and metastasis [40]. The role of Nrf2 therefore remains ambiguous, and the question of whether inhibition or induction of Nrf2 is anti-tumorigenic remains unanswered [41].

## 3. ROS and Bladder Cancer

Since elevated ROS promotes dedifferentiation of normal cells into a cancerous phenotype [8], it is not surprising that antioxidant enzymes are downregulated in bladder cancer, compared to normal bladder tissue [42]. Altering the ROS level might therefore prove to become a promising concept in treating bladder cancer [43]. It is generally accepted that the main reason for the development of bladder cancer is genotoxic insult induced by ROS. Subsequent DNA alteration then triggers malignant transformation of healthy epithelial bladder cells [44]. Genome instability and impaired DNA repair are hallmarks of carcinogenesis, driving the cells to become more sensitive to oxidative stress. This may explain why more cumulative DNA damage is found in bladder cancer patients, compared to healthy controls [45]. Bladder cancer is considered to be highly related to inflammation and hydrogen peroxide and several cytokines, including interleukin (IL) 1α, IL-6 and tumor necrosis factor-α (TNFα) accumulate during inflammation, driving bladder carcinogenesis [46]. In addition, pro-inflammatory factors such as IL-8, IL-18, cyclooxygenase-2 (COX-2), prostaglandin E2 (PGE2) and some chemokines have been associated with bladder cancer development [47]. For this reason, anti-inflammatory drugs are used to decrease the risk of developing bladder cancer [48].

Silencing the NOX4 gene in bladder cancer cells reduces intracellular ROS in vitro and blocks cell growth in vivo. Since NOX4 has been found in urothelial carcinoma and precancerous lesions but not in normal urothelium, NOX4 might be a pivotal enzyme to ROS generation, contributing to the early steps of urothelial carcinogenesis [49]. An increased ROS level seen in platinum resistant bladder cancer cells has been reported to trigger angiotensin II type 1 receptor elevation [50]. This is important, since the angiotensin II type 1 receptor both inhibits the tumor immune response [51] and encourages angiogenesis [50]. Although the mechanisms associated with ROS’s contribution to angiogenic invasiveness are not understood in detail, in vitro studies point to the relevance of ROS induced overexpression of COX-2, as well as vascular endothelial growth factor (VEGF) and HIF-1α [52]. Regardless of the exact mode of action, an increased ROS-level must be considered with acquired cisplatin-resistance.

The promotion of bladder cancer migration is also attributed to ROS. Peng et al. show that ROS targets the proto-oncogene tyrosine-protein kinase Src and the focal adhesion kinase (FAK) pathway, both of which are closely involved in motile crawling and invasive behavior of bladder cancer cells [53]. Other investigators point to pro-migratory signaling caused by an increased steady-state H_2_O_2_ level in bladder cancer cells via activation of FAK, Rho-related small GTPase Rac1 and MAPK signaling [54]. They assumed that antioxidants may provide beneficial effects in treating relapsed bladder cancer. Indeed, metastatic bladder tumor cells do have lower catalase activity, compared to non-metastatic counterparts, leading to the induction of redox-sensitive pro-tumorigenic and pro-metastatic genes, such as VEGF and matrix metalloproteinase 9 (MMP-9) [55].

### 3.1. HDAC Inhibition

Histone deacetylation (HDAC) induced by histone deacetylases (HDACs) is a major epigenetic modification seen in cancer cells. HDACs include a series of enzymes that remove acetyl groups from histones and other proteins, giving rise to chromatin remodeling and change in the gene expression profile [56]. In bladder cancer, aberrant deacetylation of histones with subsequent downregulation of tumor suppressor gene transcription is well documented [57]. Consequently, targeting HDAC and inhibiting its activity could be a promising strategy to treat bladder cancer. It is intriguing that ROS interacts with HDAC. The HDAC-inhibitor sodium butyrate has been shown to induce apoptosis in the bladder cancer cell lines, T24 and 5637, by increasing ROS production via activation of AMP-activated protein kinase (AMPK) and inactivation of the mechanistic target of rapamycin (mTOR) pathway [58]. The HDAC-inhibitor romidepsin has been shown to induce apoptosis in the J82 bladder cancer cell line. The effect was even more pronounced in H-Ras-expressing sublines, due to a higher level of ROS [59]. These in vitro findings have yet to be verified by in vivo studies. Therefore, it is uncertain if therapeutic success of an HDAC-inhibitor depends on the ROS level. This requires further investigation.

### 3.2. Nrf2-Pathway

Bladder cancer development is promoted by Nrf2. Kocanova et al. has assumed that activating the p38MAPK and PI3K signals causes upregulation of the antioxidant enzyme HO-1 through an Nrf2-mediated mechanism. There is also evidence that tumor growth is associated with somatic mutation in the Keap1-Nrf2 system, disrupting the interaction between Nrf2 and Keap1, which finally leads to constitutive stabilization and activation of Nrf2 [60,61]. Hence, Nrf2 has been closely associated with cytoprotection and anti-apoptotic properties [62]. In fact, drug-resistance of bladder cancer cells is frequently accompanied by Nrf2 overactivation [63,64]. Besides this, Nrf2 activation has also been shown to elevate HIF-1α in bladder cancer tissue [65,66]. HO-1 expression is upregulated together with HIF-1α, HIF-2α and Nrf2 in bladder cancer in comparison to healthy tissue [67]. Still, the relevance of the Nrf2-pathway in bladder cancer is not fully understood. Overexpression of Nrf2 is suggested to support the promotion and progression of cancer by suppressing oxidative stress via scavenging ROS [68]. Studies on bladder cancer cells present evidence that p62 promotes tumor cell growth by activating Keap1-Nrf2 signaling [69]. In strong contrast, p62-Keap1-Nrf2 signaling has been shown to be tumor-suppressive when elicited by ROS [70]. The relevance of Nrf2 may therefore depend on the intracellular ROS concentration, whereby high ROS and low Nrf2 levels may synergistically block tumor progression. Based on these results, attenuating Nrf2 is assumed to be beneficial in suppressing bladder cancer [71] and a variety of pharmacological approaches capitalizing on overexpression of Nrf2 are in progress. However, more experimental data are required to allow a clear assessment of Nrf2 as a therapeutic target [68].

### 3.3. MAPK Pathway

The MAPK pathway is closely involved in tumor genesis and progression. MAPK subfamilies primarily include p38 MAPK, c-Jun NH2-terminal kinase (JNK) and extracellular-signal-regulated kinase (ERK), all of which regulate cellular activities, including proliferation, differentiation, apoptosis or survival, inflammation and innate immunity [72]. Oxidative stress is one of the factors that activates MAPK [73], with an aberrant MAPK pathway thought to be significantly involved in bladder cancer [74]. When chronically exposed to arsenic, ROS induced activation of JNK and p38MAPK may be essential mechanisms for initiating bladder cancer [75]. Wang et al. argues that the carcinogenesis of bladder cancer is modulated by ROS-dependent p38 MAPK, JNK and ERK pathways since their induction results in overexpression of COX-2 [76]. Such pathway interaction has been confirmed by others. Based on TSGH-8301 bladder cancer cells, the activation of the ERK/JNK pathways has been coupled to the upregulation of COX-2 expression. COX-2 elevation is also seen when the upstream ERK activator MEK1 increases [77]. Moreover, activating the p38 MAPK pathway has been observed to induce communication with HO-1 to protect bladder cancer from apoptosis and guarantee cell survival [62].

Although these investigators point to the MAPK-pathway as a promising ROS-dependent treatment target, the role of MAPK in bladder cancer is unclear. Indeed, inhibition of proliferation and induction of apoptosis of 5637 bladder cancer cells has been shown to depend on an increased ROS level and the phosphorylation of MAPKs, including ERK, JNK and p38 MAPK [78]. Thymol-induced apoptosis of T24 cells also correlated well with the generation of ROS, JNK and p38 but not with ERK [79]. This indicates that ROS/MAPK-dependent signaling acting as an oncogenic driver is a simplification. Rather, under certain circumstances ROS and MAPK seem to reverse their oncogenic to an onco-suppressive function.

### 3.4. Nuclear Factor-KappaB (NF-κB) Pathway

The nuclear factor-kappaB (NF-κB) pathway is crucial in ROS-related mechanisms in bladder cancer, as it plays an essential role in linking inflammation to cancer. The NF-κB family contains five members designated as p65 (RelA), RelB, c-Rel, NF-κB1 and NF-κB2. In general, they are inhibited by the IκB family of proteins or by a self-inhibitory domain (NF-κB1 and NF-κB2). During chronic inflammation caused by oxidative stress, the dissociation of IκB molecules or cleavage of the inhibitory domains of NF-κB1 and NF-κB2 are necessary prerequisites to translocate NF-κB from the cytoplasm into the nucleus, thereby activating target genes to exert a pro-tumorigenic effect [80]. Under a hypoxic microenvironment, activated NF-κB, concomitant with the switch of HIF-1α to HIF-2α, forces the malignant behavior of bladder cancer T24 cells [81]. Kim et al. and others point to the critical role of ROS production (via NOX1 and NOX4) and subsequent NF-κB induction in invasive processes of bladder cancer cells in vitro and in vivo [82,83]. In contrast, suppression of NF-αB has provoked apoptosis and diminished metastatic properties of bladder cancer in vitro [84]. In a preclinical investigation NF-κB was associated with a poor prognosis, leading to the consideration that NF-κB may serve as a prognostic biomarker in bladder cancer patients [85].

### 3.5. PI3K/Akt/mTOR Pathway

The constitutive activation of the phosphatidylinositol 3-kinase/protein kinase B/mechanistic target of rapamycin (PI3K/Akt/mTOR) pathway has been documented in more than 40% of bladder cancers [86,87]. Pro-survival and drug-resistance properties are closely related to this pathway in bladder cancer [88].

The interaction between PI3K/Akt signaling and ROS, which accumulates in cancer cells, has been the subject of several investigations. In fact, ROS-dependent activation of PI3K-AKT has been observed in many cancer types [30]. Akt activation is initiated by the oxidization and inactivation of the functional antagonist phosphatase and tensin homologue (PTEN) [89]. Akt signaling promotes NADPH production, which in turn creates a situation where tumor cells adapt to the increased ROS level [90]. Inhibition of the phosphorylation levels of Akt and Akt-related signaling with subsequent generation of ROS may, therefore, be a possible cancer fighting strategy [91].

This is important, since an increased ROS level coupled to Akt pathway suppression also inhibited glucose-6-phosphate dehydrogenase (G6PD), a rate-limiting enzyme of the pentose phosphate pathway, weakening survival of bladder cancer cells [92]. Furthermore, accumulation of ROS has been proven to induce G2/M phase arrest and apoptosis via inactivation of the PI3K/Akt pathway in the bladder cancer cell line T24 [93]. Cross-talk between ROS and the PI3K/Akt/mTOR axis has meanwhile been confirmed by others as the main cause for apoptosis modulation in bladder cancer [94,95]. Zhao and coauthors have additionally pointed to combined Akt/ERK phosphorylation inhibition, triggered by excessive ROS generation, as mediators of both apoptosis and autophagy [96]. Recent investigations indicate that mTOR/ROS is not only involved in autophagy and apoptosis regulation but also controlling bladder cancer cell migration in vitro and in vivo [58]. Concerted action on ROS and mTOR/Akt may, therefore, serve to combat both growth and invasive activity of bladder cancer, but pertinent clinical trials have not yet been presented. Plumbagin, a natural plant-derived drug extracted from Chinese herbals, promotes apoptosis and stops the migration of bladder cancer cells in vitro and in vivo by ROS generation and inhibition of Akt/mTOR [97]. Gallic acid, a hydrolyzable tannin, has also been shown to inhibit T24 cell proliferation, migration and invasion, accompanied by ROS accumulation and decreased phosphorylation of PI3K and Akt [98].

Nevertheless, although the concept of combining ROS-production with Akt inhibition is attractive, the role of ROS, depending on its level, should be kept in mind. A mild-to-moderate ROS level is associated with pro-tumorigenic effects while only an excessive ROS concentration is linked to tumor-suppressive activity [99]. Therefore, the therapeutic strategy to augment ROS production might be considered for tumors with an already high ROS burden to produce selective toxicity [68]. Since elevation of ROS, along with hyperactivated Akt, is also seen in the context of adaptive processes, simultaneous targeting of ROS and Akt could be considered to re-establish chemo-/radio-sensitivity. Table 1 provides an overview of effects of ROS on bladder cancer.

## 4. SFN in Bladder Cancer

### 4.1. Preclinical and Clinical Studies

SFN’s role as a natural HDAC-inhibitor is highly relevant, since 90% of all cancers can be attributed to epigenetic modification [100,101]. Several US Food and Drug Administration (FDA) approved HDAC-inhibitors are currently in clinical use. Unfortunately, drugs such as Vorinostat, Belinostat, Panobinostat and Romidepsin are all associated with adverse reactions and resistance development [102]. These disadvantages could possibly be avoided by adding SFN to therapy [103]. Since SFN can easily be extracted from cruciferous vegetables and produced in large quantities, it could serve as a convenient and economic strategy for bladder cancer chemoprevention.

Cell proliferation inhibition, cell-cycle arrest, apoptosis induction, invasion and metastasis blockade all take place after bladder cancer cells are exposed to SFN [104]. SFN exerts stronger anti-proliferative effects on bladder cancer cell lines under hypoxia, compared to normoxic conditions [105]. This is important since hypoxia facilitates cancer progression, suggesting that SFN may be highly efficient in high grade, rapidly growing tumors where ROS is on the increase.

Growth of human bladder cancer xenografts in mice gavaged daily with SFN (52 mg/kg body weight) for 2 weeks reduced tumor weight by 42% [106] and was accompanied by downregulated HDAC activity [107]. In another study a 63% inhibition was noted when tumor bearing mice were treated with SFN (12 mg/kg body weight) for 5 weeks. The SFN extract changed the morphological appearance of tumors, with increased karyopyknosis and decreased angiogenesis [108]. In both investigations, the administration of SFN did not evoke apparent toxicity [106,108]. There is also evidence that SFN may protect against chemical-induced bladder cancer by normalizing the composition of gut microbiota and repairing pathophysiological destruction of the gut barrier, as well as decreasing inflammation with the accompanying immune response [109]. Interestingly, the tissue uptake level of SFN evaluated in a rat model was highest in the bladder and the stomach, whereas the tissue level of SFN in the colon, prostate and several other organs was very low [110]. These differences indicate that SFN could be particularly effective in treating bladder cancer.

The relevance of SFN in treating patients with bladder cancer has not yet been clarified. A prospective study involving nearly 50,000 men indicated that high cruciferous vegetable consumption may reduce bladder cancer risk [111]. A strong and significant inverse association was observed between bladder cancer mortality and broccoli intake in a case-control study including 239 bladder cancer patients, with a significant reduction of disease-specific death (57% reduction) and overall mortality (43% reduction) [112]. Tang et al. concluded, from their study conducted on 275 individuals with cancer and 825 individuals without cancer, that cruciferous vegetables, when consumed raw, may reduce the risk of bladder cancer [113]. Finally, reviewing all meta-analyses on modifiable risk factors for primary bladder cancer, the intake of cruciferous vegetables has been demonstrated to be chemopreventive [114].

### 4.2. SFN’s Influence on ROS

Evidence shows that SFN upregulates the ROS level in T24 bladder cancer cells to induce apoptosis [115]. SFN treatment has been associated with loss of the mitochondrial membrane potential, with cytochrome c release and alteration of the Bcl-2/Bax ratio. In addition, SFN increases the activity of caspase-9 and -3, but not of caspase-8, and mediates the cleavage of poly ADP-ribose polymerase (PARP). These findings have been interpreted such that SFN triggered ROS generation modifies the intrinsic apoptotic pathway [115]. Other investigators have shown that SFN-induced apoptosis of 5637 cells via a ROS-dependent pathway is linked to both caspase-8 and -9 activation, indicating an influence of both intrinsic and extrinsic apoptotic pathways [116]. Therefore, it seems that ROS exerts its effects on apoptosis via different mechanisms, depending on the cell line. This means that depending on the cancer type, SFN may initiate apoptotic events in bladder cancer patients over different mechanisms and to a differing extent. It would be of interest to explore whether high and low responders to SFN can be differentiated.

Jin et al. have shown that SFN-induced ROS generation promotes tumor necrosis factor-related apoptosis-inducing ligand (TRAIL) sensitivity [71]. Application of SFN to TRAIL-resistant bladder cancer cell lines has resulted in the truncation of the pro-apoptotic protein Bid and induction of the death receptor 5 (DR5), which finally led to cell death. Blockade of ROS generation inhibited apoptotic activity and prevented Nrf2 activation in cells treated with SFN, pointing to a direct effect of ROS on apoptosis. Since SFN combined with TRAIL upregulated ROS generation and reduced Nrf2, these investigators hypothesized that SFN sensitizes the tumor cells to TRAIL-mediated apoptosis by generating ROS that overcomes an Nrf2-mediated defense system established in resistant cells [71].

Meanwhile, the therapeutic options for bladder cancer have been considerably widened by the introduction of immune checkpoint inhibitors targeting cytotoxic T lymphocyte antigen 4 (CTLA4) and programmed death (ligand) 1 (PD-1/PDL1). In view of these new treatment modalities aimed at suppressing tumor cell inhibition of anticancer immune responses [117], it has become evident that SFN may exert immunomodulatory effects by altering the proportion of natural killer cells, monocytes, dendritic cells, T- and B-cells [118]. Investigation of human peripheral blood T-cells derived from healthy donors showed that SFN induces a significant accumulation of ROS in the T-cells, which resulted in the inhibition of T-cell activation and T-cell effector functions [119]. If a bladder cancer patient has been subjected to immunotherapy, the additional use of SFN may, therefore, bear the risk of blocking the T-cell-mediated immune response [120].

### 4.3. SFN Acts on Nrf2

SFN potently inhibits carcinogenesis via activation of the Nrf2 pathway [121]. The daily administration of an aqueous extract of broccoli sprouts to rats (equivalent to isothiocyanate doses of 40 μmol/kg and 160 μmol/kg body weight) inhibited *N*-butyl-*N*-(4-hydroxybutyl) nitrosamine induced bladder cancer development and was associated with a significant induction of glutathione S-transferase and NAD(P)H:quinone oxidoreductase 1 [122]. Induction of these enzymes was largely mediated by Nrf2 [123]. Notably, Nrf2 activation by SFN in the bladder occurred primarily in the epithelium, which is the principal site of bladder cancer development. Since Nrf2 is critical to stimulating a variety of cytoprotective genes and is closely involved in inhibiting DNA damage, activating Nrf2 by SFN might be a key strategy to prevent bladder cancer initiation [124]. Still, the relevance of the Nrf2 pathway for bladder cancer progression is not completely understood. In fact, Nrf2 induction has also been considered a secondary process, following an increased ROS level and endoplasmic reticulum stress evoked by SFN [125]. In this case, Nrf2 could exert a prosurvival role by hindering ROS-induced apoptosis [71], and the overexpression of Nrf2 target genes could support cell proliferation by increasing ribonucleotide synthesis, serine biosynthesis and autophagy [68]. Recently, a hormetic action was found in an angiogenesis assay where 2.5 µM SFN promoted endothelial tube formation but inhibited it at 10–20 µM [126]. Whether the dose-dependency seen with SFN contributes to the role of Nrf2 as an oncoprotein or a tumor suppressor remains open.

Nrf2 may exhibit opposing functions, whereby chemopreventive mechanisms should be differentiated from tumor-suppressive effects [68]. Evaluation of bladder chemoprevention in a murine model revealed that Nrf2 knockout was associated with a higher incidence of bladder cancer [127]. Upregulation of Nrf2 was beneficial in protecting normal epithelial cells against cancer development by increasing metabolism of carcinogenic compounds. In contrast, elevation of Nrf2 seen in cancer cells, drives tumor progression and contributes to drug resistance. Therefore, to counteract tumor progression it is necessary to interrupt Nrf2 signaling, once normal epithelial cells have been converted to a malignant phenotype.

Whether SFN acts on Nrf2 in a biphasic manner is not clear. Studies in regard to bladder cancer reviewed here are related to chemoprotective effects of SFN. No publications are available dealing with the influence of SFN on Nrf2 in bladder cancer cells (where Nrf2 downregulation would be expected). SFN activated Nrf2 in normal esophageal epithelial cells, but not in esophageal cancer cells [128], which seems to be in accordance with the biphasic characteristics of SFN. However, inhibition of cell proliferation and invasion of pancreatic cancer cell lines by SFN was associated with translocation of Nrf2 from the cytoplasm into the nucleus [129]. Based on colon cancer cell lines, tumor suppressive effects of SFN were also paralleled by nuclear Nrf2 accumulation [130]. Certainly, the relevance of SFN’s effects on Nrf2 signaling requires further analysis. Whether Nrf2 is specifically targeted by SFN is not clear. Experiments on immortalized kidney epithelial cells demonstrated the association between apoptotic events, reduction of Nrf2 and Akt signaling, with Akt but not Nrf2 being the major apoptosis regulator [131]. Kombairaju et al. even concluded from their studies on lung cancer cells that SFN induced Nrf2 signaling might be an epi-phenomenon, not relevant for tumor growth regulation [132].

### 4.4. SFN and MAPK Signaling

Although the influence of SFN on the MAPK pathway has been documented, respective experiments on bladder cancer cells are sparse. SFN upregulates the expression of two Nrf2-dependent enzymes, glutathione transferase (GSTA1-1) and thioredoxin reductase (TR-1), and downregulates COX-2 in T24 cells, which is closely associated with p38 MAPK activity [133]. Abbaoui and coworkers observed apoptosis and tumor weight reduction in murine UMUC3 xenografts exposed to SFN. The antitumor effect of SFN was associated with downregulation of both the epidermal growth factor receptor (EGFR) and the human epidermal growth factor receptor 2 (HER2/neu) [106]. This is remarkable, since inhibition of either EGFR or HER2 signaling has been shown to correlate with enhanced p38 MAPK phosphorylation [134]. Gemcitabine or cisplatin treatment in human bladder cancer models has been shown to cause a dose-dependent release of ROS and activate the p38 MAPK-signaling pathway [135]. The similarity between gemcitabine/cisplatin and SFN triggered pathway alterations in bladder cancer may open new therapeutic strategies, including a combined treatment regimen to cause additive effects. SFN may also serve as an alternative drug candidate, once gemcitabine/cisplatin resistance has occurred.

Data on MAPK signaling in bladder cancer are sparse and do not allow final conclusions as to how MAPK may communicate with ROS and Nrf2. Concerning the bladder cancer cell line T24, MAPK was shown to act as an upstream mediator of Nrf2 [133]. Inhibition of colorectal cancer cell proliferation and migration by SFN was induced by phosphorylating MAPK, which subsequently promoted Nrf2 accumulation and—in parallel—enhanced ROS [130]. Still, similar experiments on colorectal tumor cells also documented that MAPK may directly depend on ROS, serving as the key factor to modulate tumor growth [136]. The close dependence of MAPK-signaling pathways on ROS has also been shown for thyroid cancer cells [137]. Thus, the interaction between ROS/Nrf2 and MAPK is complex and can occur in a two-sided direction with either ROS or MAPK as the initial stimulus.

### 4.5. SFN and NF-κB Signaling

NF-κB signaling correlates with aggressive bladder cancer behavior and poor clinical outcome [138]. Therefore, NF-κB inhibitors have been proposed as efficacious targeted therapies [139]. Concomitant NF-κB inhibition has been observed in BIU87 bladder cells, as SFN inhibits cell proliferation, arrests the cell cycle at the G2/M phase and induces apoptosis [140]. Although the underlying mode of action has not been explored in detail, the authors suggest that the insulin-like growth factor-binding protein-3 (IGFBP-3) is critically involved in suppressing NF-κB, either by blocking IGF1 signaling, by acting on cell-cycle-regulating proteins or by interfering with the MAPK-signaling pathway. SFN has been shown to downregulate COX-2 expression in T24 bladder cancer cells at both the transcriptional and translational level. This may be due to the nuclear translocation of NF-κB and reduced binding to the COX-2 promotor, initiated by upregulation of MAPK [141]. Further publication concerning SFN’s influence on the NF-κB pathway in bladder cancer is not available, so that the question of whether NF-κB inhibition is responsible for SFN’s chemopreventive and antitumor properties remains unanswered.

NF-κB activation occurs via two major signaling pathways: canonical and non-canonical, in response to stimuli from various cell surface receptors. Therefore, NF-κB must not always be linked to ROS. Nevertheless, exposing leukemia cells to SFN led to both enhanced ROS signaling and diminished NF-κB activity [142]. A direct interaction between ROS/Nrf2 and NF-κB in epithelial cancer cells has been reported [143].

### 4.6. SFN and Akt/mTOR Modulation

The Akt-mTOR-pathway serves as a central regulator of cell growth and proliferation. In three bladder cancer cell lines (RT112, UMUC3 and TCCSUP), SFN treatment significantly suppressed the amount of phosphorylated Akt and phosphorylation of the mTOR subunit Rictor [144]. Reduction of Akt and mTOR phosphorylation, along with diminished p70S6k downstream signaling under SFN, has also been observed in HTB-9 cells [145], pointing to a common mechanism of SFN action. The relevance of SFN as a cell-cycle inhibitor has furthermore been proven in terms of diminished expression of the cell-cycle-regulating proteins of the cyclin and CDK family. Accordingly, the CDK inhibitors, p19 and p21, are elevated under SFN [144,145]. The suppressive effect of SFN on Akt-mTOR signaling has also been seen with long-term treatment, in contrast to resistance induction evoked under chronic use of the established mTOR-inhibitors, everolimus and temsirolimus [144,146]. Further investigation into adding SFN to everolimus/temsirolimus treatment for reversion or prevention of drug resistance might, therefore, be warranted. Although the Akt/mTOR pathway is connected to ROS dependent signaling, the concerted action of SFN on Akt/mTOR–ROS has not been proven. The natural compound luteolin with strong antioxidative properties has been shown to inhibit cell survival and induce G2/M cell-cycle arrest of T24 cells. This was coupled to p21 upregulation and p70S6k downregulation [147]. A similar mode of action may hold true for SFN as well. In fact, induction of apoptosis and senescence of esophageal squamous cell carcinoma cells by SFN was triggered by a ROS-mediated mTOR inactivation [128].

**Table 1 ijms-22-05938-t001:** Effects of ROS on bladder cancer ^#^.

Cell Lines	Mechanism and Outcome	Reference
UROtsa	DNA damage ↑ Malignant transformation ↑ PARP-1 ↓	[44]
MYP3	Genetic change ↑ IL-1α/IL-6/TNF-α ↑ Carcinogenesis ↑	[46]
T24, UMUC6, KK47	NOX4 ↑ p16 ↓ Cell cycle arrest ↓	[49]
T24, 5637	AT1R ↑ VEGF ↑ Angiogenesis ↑ Platinum resistance ↑	[50]
SV-HUC-1	COX-2 ↑ VEGF/HIF-1α↑, MAPK/PI3K/AKT ↑ Angiogenesis ↑	[52]
TSGH-8301 ^##^	Src ↑ FAK ↑ Migration ↑ EMT ↑ Stress fibers ↑	[53]
253J, 253J-BV	p130Cas ↑ FAK ↑ Rac1 ↑ MAPK ↑ Invasion ↑	[54]
253J, 253J-BV	MMP-9 ↑ VEGF ↑ Metastasis ↑	[55]
T24, 5637	cleaved Caspase 3 ↑ PARP ↑ AMPK ↑ Autophagy/Apoptosis ↑ MMP ↓ Bcl-2/Bax ↓ mTOR ↓	[58]
J82, J82-Ras	Caspase 3,7 ↑ Apoptosis ↑ MMP ↓ GSH ↓	[59]
T24, 5637	p62/Nrf2 ↑ Keap-1 ↓	[70]
TRAIL resistant	TRAIL ↑ DR5 ↑ Apoptosis ↑ Bid ↓ Nrf2 ↓ T24, J82, HT1376	[71]
SV-HUC-1	ATF2 ↑ JNK/p38MAPK ↑ Carcinogenesis ↑	[75]
SV-HUC-1	JNK/ERK/p38MAPK ↑ COX-2 ↑ Carcinogenesis ↑	[76]
TSGH-8301	JNK/ERK ↑ AP-1 ↑ COX-2 ↑ Carcinogenesis ↑	[77]
5637	JNK/ERK/p38MAPK ↑ activated caspase 3, 9 ↑ Bcl-2/Bax ↓	[78]
T24	JNK/p38MAPK ↑ cleaved Caspase 3, 9 ↑ Proliferation ↓ Apoptosis ↑	[79]
253J-BV	NF-κB ↑ Invasiveness↑	[82]
MB-49, SV-HUC-1	NF-κB ↑ Bcl-2/Bax ↑ Proliferation ↑ Apoptosis ↓	[83]
T24	p21WAF1/CIP1 ↑ cleaved Caspase 3, 8, 9 ↑ PARP ↑ Apoptosis ↑ CyclinA/Cyclin B1 ↓ Bcl-2/Bax ↓ PI3K/AKT ↓ Cell Cycle ↓	[93]
T24	LC3-II ↑ cleaved Caspase 3, 8 ↑ Autophagy/Apoptosis ↑ AKT/ERK ↓ Bcl-2/Bax ↓	[96]
T24	cleaved Caspase 3 ↑ p53 ↑ Cyt-c ↑ Apoptosis ↑ Bcl-2/Bax ↓ MMP ↓ PI3K/Akt/NF-κB ↓	[98]
5637	CyclinB1 ↑ pCDK1 ↑ activated Caspase 3, 8, 9 ↑ Cleaved PARP ↑ Mitotic arrest/Apoptosis ↑	[116]
T24	cleaved Caspase 3, 9 ↑ PARP ↑ Cyt-c ↑ Nrf2/HO-1 ↑ Apoptosis ↑ Bcl-2/Bax ↓ MMP ↓ Cell growth ↓	[125]
T24	DNA damage ↑ p38MAPK ↑ Apoptosis ↑ Proliferation ↓	[135]
T24	mTOR, p16 ↑ p21 ↓ Proliferation ↑ Cell cycle arrest ↓	[147]

^#^ Effects are all related to ROS-elevation. ↑: Up-regulation, ↓: Down-regulation. No information was available about the degree of elevation. ^##^ Probably contaminated (https://web.expasy.org/cellosaurus/CVCL_A342. accessed on 25 May 2021).

Investigation employing a collagen gel culture model has indicated a potential link between Akt and integrin α and β subtypes [148]. This is notable, since integrins are important drivers of cell adhesion and migration. A connection between integrins and Akt may explain why SFN not only diminishes bladder cancer cell adhesion but migration as well. In fact, the process of migration regulation is accompanied by altered α and β expression levels [149]. Modification of redox homeostasis by inducing the production of radicals and mitochondrial depolarization has been associated with overexpression of α5 and β1 integrin subunits, providing cross-communication between integrin adhesion receptors and oxidative stress, presumably via ROS [150].

In addition, CD44, a non-kinase transmembrane glycoprotein, has been found to be modified by SFN in the adhesion and migration model. CD44 exists as the CD44 standard (CD44s) form and specific CD44 variant splice (CD44v) isoform. CD44 is not only involved in cytoskeletal changes and cellular motility but also serves as a cancer stem cell (CSC) marker [151,152]. In breast cancer, CD44v has been shown to inversely correlate with CSC signatures, and switching from CD44s to CD44v was essential for cells to undergo mesenchymal–epithelial transition [151]. An increased expression level of the CD44 variants v3–v7 on bladder cancer cells following SFN exposure was recently documented by Justin et al. [149]. Hypothetically, CD44v elevation caused by SFN forces the acquisition of an epithelial phenotype and prevents adaptive plasticity of bladder cancer cells. Whether SFN diminishes CD44s in parallel and whether SFN action on CD44 correlates with oxidative stress requires further investigation. Sasaki’s work has indicated a close interaction between CD44s and HIF1α-mediated resistance to oxidative stress [153], and Xu et al. have shown that reduced CD44s expression correlates with induced intracellular levels of ROS in bladder cancer cells [154]. It seems likely, therefore, that SFN’s action on CD44 expression is linked to ROS and ROS-related pathways.

## 5. Conclusions

The natural HDAC-inhibitor, SFN, acts in a multifaceted fashion on bladder cancer, leading to cell growth arrest, proliferation blockade, apoptosis induction, along with suppression of tumor cell motility and invasion. SFN’s inhibitory activity is not restricted to bladder cancer but is apparent in other tumor types as well. Apoptosis induction by SFN via ROS is seen in hepatocellular carcinoma [155], lung cancer [156] and breast cancer cells [157]. How SFN specifically targets bladder cancer remains to be clarified. Several molecular pathways associated with bladder cancer could serve as potential targets. These include CD44-related signaling [158], Notch and MAPK signaling [159], Akt/mTOR signaling [160] or JAK/STAT and NF-κB/Snail signaling [161]. SFN influences all of these signaling pathways, making it an interesting candidate for supportive tumor therapy. Notably, negative side-effects and resistance-induction, as encountered with established drugs, are not evoked by SFN, which could further strengthen its clinical usefulness. Still, the relevance of ROS and ROS-related pathways for bladder cancer progression is not fully elucidated and both tumor-promoting and tumor-suppressing activities have been documented.

Based on current knowledge, mild elevation of ROS activates pro-tumorigenic survival and tumor growth, whereas excessive concentration of ROS leads to the induction of cell death. Thus, opposing strategies must be critically evaluated. This includes either therapeutic downregulation of ROS to prevent oncogenic signaling or upregulation of ROS above a sensitive threshold to cause oxidative damage (Figure 2).

Ongoing studies are required to precisely define the role of ROS on tumorigenesis and cancer progression. Accordingly, the consequences of SFN–ROS communication in regard to tumor cell behavior should be explored in more detail. In particular, SFN’s mode of action in tumor cells with a moderate versus substantial ROS level should be evaluated. Since the response of tumor cells to radiotherapy or chemotherapy is promoted by increased ROS production, ROS inhibition may at least be partially responsible for therapeutic resistance. In this context, treatment targeting the antioxidative stress system is an important research direction to counteract radioresistance and chemoresistance. Intriguingly, SFN has overcome cisplatin-based resistance via ROS-modulation. This is highly relevant in regard to second line treatment options. Further investigation is essential to determine the degree to which ROS contributes to the development of resistance processes triggered by undesired feedback loops and to what degree SFN counteracts tumor cell re-activation in the course of chemotherapy. Finally, SFN’s considerable antitumor potential has been documented in vitro and in vivo but not in tumor patients. Limited bioavailability of SFN remains a hurdle, necessitating further investigation into increasing bioavailability. Genetically altered plants with significantly higher amounts of glucoraphanin have been developed, which might overcome this problem. Nano-encapsulation and the synthesis of potent SFN analogues may also increase the bioavailability of SFN. Therefore, many aspects regarding SFN application remain to be investigated before a final conclusion can be drawn in respect to its use as an anticancer compound.

## Figures and Tables

**Figure 1 ijms-22-05938-f001:**
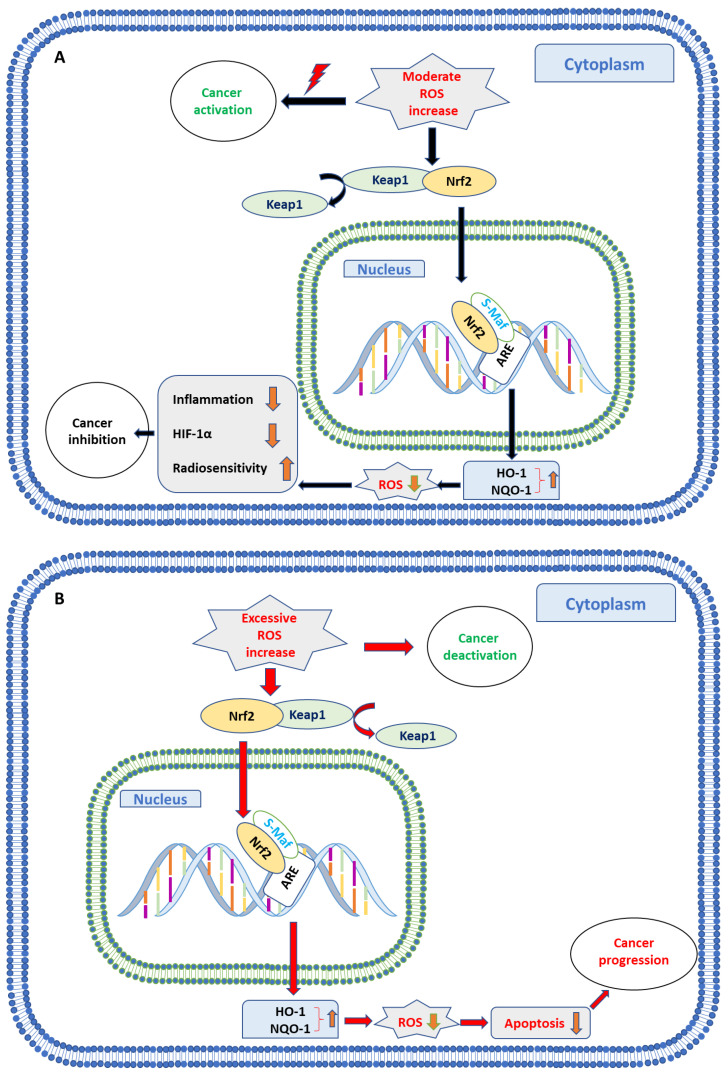
Double role of ROS in cancer cells. Increased ROS dissociates Keap1 and Nrf2, resulting in the translocation of Nrf2 and s-Maf to ARE in the nucleus which activates antioxidant enzymes such as HO-1 and NQO-1. Initial moderate ROS increase (**A**) is related to cancer development and progression, whereby elevation of HO-1 and NQO-1 downregulates ROS, subsequently blocks inflammatory processes, inactivates HIF-1α and induces radiosensitization. An initial excessive ROS level (**B**) causes toxic effects and apoptosis, whereby a decrease of ROS by HO-1 and NQO-1 protects cancer cells from oxidative toxicity, enhancing survival and invasive capability.

**Figure 2 ijms-22-05938-f002:**
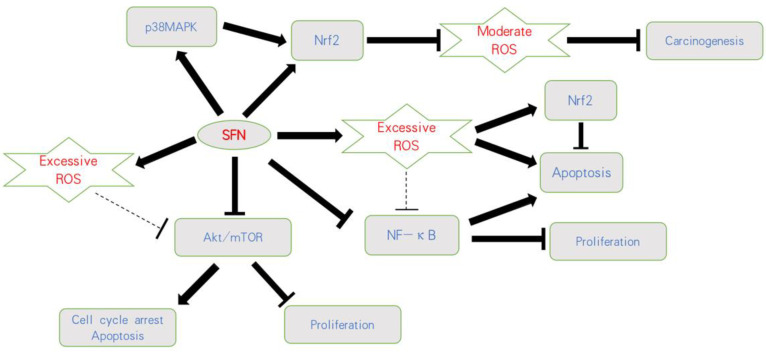
Influence of SFN on ROS-related pathways on bladder cancer. SFN blocks carcinogenesis by activating Nrf2 or the p38MAPK/Nrf2 axis and counteracting a moderate ROS-increase. Based on an initially excessive ROS level, SFN further increases ROS, resulting in apoptosis and proliferative inhibition. Nrf2 is thus considered a secondary product, followed by a ROS-increase involving anti-apoptotic properties. SFN also acts on the Akt/mTOR and NF-κB pathways, whereby the relevance of ROS as a trigger factor has not finally been validated; 
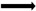
 indicates activation; 
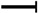
 indicates inhibition; 
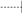
 indicates not clear.

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
