# Peer review of "Sulforaphane Impact on Reactive Oxygen Species (ROS) in Bladder Carcinoma"

_ijms, 2021, doi:10.3390/ijms22115938_

Round 1

Reviewer 1 Report

Manuscript entitled "Sulforaphane impact on reactive oxygen species (ROS) in bladder carcinoma". Overall this work is not comprehensive. It is not so relevant, neither.

Major issues:

  1. The authors should provide review on the clinical impacts of ROS in UC.
  2. The authors should provide therapeutic relevance of ROS in UC in more detail.
  3. The table and figure is not well formatted.

Minor: TSGH8301 is not a real cell line, please take down the description related to Ref. 74.

Author Response

Comment 1: The authors should provide review on the clinical impacts of ROS in UC. The authors should provide therapeutic relevance of ROS in UC in more detail.

Our answer: The role of ROS in cancer is not clear yet. Therefore, we cannot make final conclusions as to the role of ROS as a treatment target. Nevertheless, since ROS is altered in tumor cells and since SFN is assumed to exert anti-tumor effects, it is worthwhile to explore how SFN influences ROS and how this interaction may contribute to its anti-tumor potential. In this context, the main focus was laid on SFN and its influence on ROS. Interestingly, the influence is not clearcut. Rather pros and cons in regard to ROS levels have to be considered, which led to the final conclusion that “many aspects regarding SFN application remain to be investigated”. Meanwhile, a review article has been published documenting that ROS in fact may provide therapeutic relevance in treating bladder cancer (Liu et al. Clin Transl Oncol. 2020;22:1687-1697). It is noted by the authors that “redox balance has gained interest for its prospective application for clinical trials and experiments in animal models” and “changes in ROS levels and an oxidative stress state cast new light on the treatment of bladder cancer”. We have now included this article in the manuscript which now reads: “Since elevated ROS promotes dedifferentiation of normal cells into a cancerous phenotype [8], it is not surprising that antioxidant enzymes are down-regulated in bladder cancer, compared to normal bladder tissue [42]. (line 145)Altering the ROS level might therefore prove to become a promising concept in treating bladder cancer [43].”.

Comment 2. The table and figure is not well formatted.

Our answer: We have now better explained the table and included arrows indicating the direction of change for each observed parameter in the different cell lines. Figure 1 has been divided into two sub-figures to better demonstrate the dual role of ROS.

Comment 3. TSGH8301 is not a real cell line, please take down the description related to Ref. 74.

Our answer: We are thankful for this important comment. The cell line TSGH8301 was established in 1988 (Yeh et al. J Surg Oncol. 1988;37:177-84). Since then over 50 manuscripts have been published dealing with it. Still, it is correct that TSGH8301 is problematic due to a contamination with the cervix cancer cell line ME-180 (https://web.expasy.org/cellosaurus/CVCL_A342). We have now informed the readership that TSGH8301 might be contaminated and added this as a footnote to table 1 (line 347): “probably contaminated (https://web.expasy.org/cellosaurus/CVCL_A342)”.  

Reviewer 2 Report

In the revised manuscript, the authors’ points are clear and supported by newly added reference publications, which will clarify readers’ understanding. I recommend the manuscript for publication after the following minor revisions.

Major comments:

  • It is assumed that the illustration in Figure 1 has two components: the cancer inhibition effects and the cancer progression effects by Nrf2 according to the cellular ROS levels. However, the current schema could mislead the readers. To date, genetic studies have clearly established that the heterodimer, Nrf2 and s-MAF (a transcription factor belonging to the MAF family) interacts with the antioxidant response element (ARE) and activates the expression of cytoprotective genes; however, it is highly unlikely that two molecules of Nrf2 bind to one ARE as described in the schema of Figure 1. The authors need to modify Figure 1. It is highly recommended to indicate two separate illustrations showing the cancer cells harboring two different levels of ROS (i.e. moderate ROS level or excessive ROS level), as well as the correspondent effects of Nrf2. Also note that the Keltch-like ECH-associated protein 1 as described as ‘Keap-1’ in Figure 1 is not a common notation, which also leads to the inconsistent use of notation shown in the main text as ‘Keap1’. The authors should do the simple editing and change Keap-1 to Keap1 in Figure 1.
  • The authors summarize the reported effects of ROS on the bladder cancer cell lines in Table 1, titled Effects of ROS on bladder cancer. However, it is not clear whether the listed publications studied a moderate ROS accumulation or an excessive ROS accumulation in cancer cell lines. As mentioned in the manuscript (Page 132, line 102), ROS seems to display both pro- and anti-tumorigenic properties according to the intracellular ROS levels, which is assumed to be tightly associated with the mechanisms and outcomes reported in the listed publications. Thus, it is important to state clearly the degree and direction of ROS levels studied in each publication in Table 1.

Minor comments:

  • To help readers’ understanding, it should be worth mentioning that a high level of Nrf2 activation in various cancer types as mentioned on Page 3, line 116, is often associated with the somatic mutations in the Keap1-Nrf2 system, which disrupts the interactions between Nrf2 and Keap1, resulting in constitutive stabilization and activation of Nrf2. Indeed, bladder cancer is one of the well-known solid tumors that Keap1 and Nrf2 are frequently mutated when compared to other solid cancers (Kerins et al., Sci Rep, 2017, Wu et al., Rev. Cancer Biol., 2020). As this is an important aspect that needs to be considered in the context of clinical intervention with Sulforaphane, it is highly recommended to be included in the manuscript.

Author Response

Comment 1: It is assumed that the illustration in Figure 1 has two components: the cancer inhibition effects and the cancer progression effects by Nrf2 according to the cellular ROS levels. However, the current schema could mislead the readers. To date, genetic studies have clearly established that the heterodimer, Nrf2 and s-MAF (a transcription factor belonging to the MAF family) interacts with the antioxidant response element (ARE) and activates the expression of cytoprotective genes; however, it is highly unlikely that two molecules of Nrf2 bind to one ARE as described in the schema of Figure 1. The authors need to modify Figure 1. It is highly recommended to indicate two separate illustrations showing the cancer cells harboring two different levels of ROS (i.e. moderate ROS level or excessive ROS level), as well as the correspondent effects of Nrf2. Also note that the Keltch-like ECH-associated protein 1 as described as ‘Keap-1’ in Figure 1 is not a common notation, which also leads to the inconsistent use of notation shown in the main text as ‘Keap1’. The authors should do the simple editing and change Keap-1 to Keap1 in Figure 1.

Our answer: As suggested, we have edited appropriately, changing Keap-1 to Keap1. Figure 1 has been separated into two separate figures, figure 1A+B, and modified with regard to Nrf2 and ARE. s-Maf has been included in the figure and is explained in the text, which now reads (line 112): “Binding of Nrf2 along with the transcription factor s-Maf to ARE ….”..

Comment 2: The authors summarize the reported effects of ROS on the bladder cancer cell lines in Table 1, titled Effects of ROS on bladder cancer. However, it is not clear whether the listed publications studied a moderate ROS accumulation or an excessive ROS accumulation in cancer cell lines. As mentioned in the manuscript (Page 132, line 102), ROS seems to display both pro- and anti-tumorigenic properties according to the intracellular ROS levels, which is assumed to be tightly associated with the mechanisms and outcomes reported in the listed publications. Thus, it is important to state clearly the degree and direction of ROS levels studied in each publication in Table 1.

Our answer: The modifications listed are all related to ROS-elevation. Unfortunately, no details are provided in the cited literature about the degree of elevation. Therefore, we cannot provide further information concerning this aspect. A corresponding footnote has now been included which reads (line 346): “Effects are all related to ROS-elevation. No information was available about the degree of elevation”.

Comment 3: To help readers’ understanding, it should be worth mentioning that a high level of Nrf2 activation in various cancer types as mentioned on Page 3, line 116, is often associated with the somatic mutations in the Keap1-Nrf2 system, which disrupts the interactions between Nrf2 and Keap1, resulting in constitutive stabilization and activation of Nrf2. Indeed, bladder cancer is one of the well-known solid tumors that Keap1 and Nrf2 are frequently mutated when compared to other solid cancers (Kerins et al., Sci Rep, 2017, Wu et al., Rev. Cancer Biol., 2020). As this is an important aspect that needs to be considered in the context of clinical intervention with Sulforaphane, it is highly recommended to be included in the manuscript.

Our answer: This aspect has now been included. Section 3.2 now reads: “Bladder cancer development is promoted by Nrf2. Kocanova et al. has assumed that activating the p38MAPK and PI3K signals causes up-regulation of the antioxidant enzyme HO-1 through an Nrf2-mediated mechanism. (line 200) There is also evidence that tumor growth is associated with somatic mutation in the Keap1-Nrf2 system, disrupting the interaction between Nrf2 and Keap1, which finally leads to constitutive stabilization and activation of Nrf2 [60,61].”.

Reviewer 3 Report

The paper is suitable for the publication

Author Response

N/A

Round 2

Reviewer 1 Report

nil

This manuscript is a resubmission of an earlier submission. The following is a list of the peer review reports and author responses from that submission.

Round 1

Reviewer 1 Report

Manuscript entitled "Sulforaphane impact on reactive oxygen species (ROS) in bladder carcinoma"

This work is limited by the following issues:

  1. Sulforaphane is not well-documented in UC treatment. A review on this topic would be of very limited impact.
  2. ROS is an important biological issue in cancer yet not mature enough for cancer treatment. Thus a review on Sulforaphane impacts ROS in UC might not be interesting to general readers.
  3. The figures is not well-prepared.

Reviewer 2 Report

In this manuscript, the authors described the anti-cancer effects of sulforaphane (SFN) in the treatment of bladder cancer. The mechanism of action seems to be mediated by ROS. The review is well written but requires major improvements before publication.

ROS and cancer. This paragraph describes only in part the complexity of this story. This section should be further implemented. 

I suggest adding a table that summarizes the alterations of ROS pathways in bladder cancer and the SFN’s influence on these.

I wonder if the anti-cancer action of Sulforaphane is specific for bladder cancer (BC) or in common with other tumor types. I guess if it is possible to identify some SFN tumour-specific target that is characteristic of BC.

Supplementary Materials: The following are available online at www.mdpi.com/xxx/s1, Figure S1: 455 title, Table S1: title, Video S1. It is not clear if Suppl. Materials exist or not. If not, I suggest removing this sentence. If yes, the link does not work.

Reviewer 3 Report

In this manuscript, the authors summarize the potential role of sulfuraphane (SFN), a phytochemical found in cruciferous plants such as broccoli, for a treatment for bladder carcinoma. It has been reported that SNF shows therapeutic effects for several diseases including inflammatory diseases, diabetes, autism and cancers in vivoand/or in vitrostudies, as well as in clinical studies. The signaling pathway and intracellular targets of SFN mediating these therapeutic effects have been elucidated since the early 1990s, indicating its involvement in a wide range of molecular mechanisms. In the manuscript, the authors focus particularly on the reactive oxygen species (ROS) related biological process during the course of cancer development, which is promoted as one of the targeting pathways of SFN treatment for bladder carcinoma.

A major caveat regarding the current manuscript is that the main subject of this review article is not fully addressed. The listed items need to be improved to meet the standard of quality for publication.

Major comments:

Although the manuscript states that the current review deals with the question of how SFN may modulate ROS in bladder cancer and how intervention into ROS related signaling by SFN administration may provide innovative anti-tumor strategies, the indicated main subject is not fully supported in the current manuscript. The potential of SFN in bladder cancer treatment, including possible molecular mechanisms, is well described in the manuscript. Also, the ROS mediated pathways in molecular cancer pathogenesis and chemoprotection are well summarized independently from SFN treatment. However, the key relevance between the therapeutic effect of SNF and the ROS dependent mechanism in bladder cancer treatment is not substantially supported by reference articles.

  • The authors need to emphasize paragraph 4.6 titled, “SFN’s influence on ROS”. In Chapters 3 and 4, the authors summarize the signaling pathways in bladder cancer from two different aspects; (1) the ROS related molecular events and pathways during the process of cancer development, (2) the targeting pathway by SFN mediating anti-tumorigenic effects in bladder cancer. However, it is not clear whether chemoprotective effects by SFN administration are directly supported by ROS-mediated mechanism either through HDAC inhibition, Nrf2 signaling, MAPK pathway, NF-kb pathway and Akt/mTOR pathway. Indeed, reference 107 is the only article showing direct molecular correlation between the change of ROS level in the bladder cancer cell line by SFN and Nrf2 signaling in the section 4.1 – 4.5. Consequently, paragraph 4.6 should be emphasized in the manuscript, because the authors seem to address the direct interaction of how SFN may contribute to ROS-mediated anti-tumorigenic effects in bladder cancer in this paragraph. Thus, the authors need to provide stronger evidences (reference articles) and a more informative discussion, particularly in paragraph 4.6.
  • As described above, no clear evidence is shown (or does not exist) to support the authors’ point that SFN may provide chemoprotective effects in bladder cancer via the ROS mediated mechanism. To help readers’ understanding, a table that summarizes the ROS-mediated pathways in bladder cancer pathogenesis and/or chemoprotection, and the potential target pathway of SFN in bladder cancer treatment in a corresponding manner should be necesarry. The table also needs to include reference articles, respectively.

Minor comments:

  • Page 1, line 41. SFN is not an herbal drug.
  • Figure 1. ARE (antioxidant response element) is the specific biding site of DNA. However, the current schema indicating ARE looks like some kind of co-factors mediated with Nrf2 transcription. Please modify the schema of ARE.
  • Page 2, line 50. It is described that ”it has recently been discovered that SFN may act as a redox regulator” by indicating reference article 8. However, the critical contribution of SFN to a redox system by activating Nrf2, a redox regulator, was reported over 10 years ago (Dinkova-Kostova et al., 2002), which is not a ‘recent’ discovery.